# Characteristics and Outcomes of Diffuse Interstitial Pneumonias Discovered in the ICU: A Retrospective Monocentric Study—The “IPIC” (Interstitial Pneumonia in Intensive Care) Study

**DOI:** 10.3390/diagnostics15161995

**Published:** 2025-08-09

**Authors:** Damien Eckert, Julien Bermudez, Marc Leone, Mathieu Di Bisceglie, Florent Montini

**Affiliations:** 1Service d’Anesthésie et de Réanimation, Hôpital Nord, APHM, 13015 Marseille, France; damien.eckert@ap-hm.fr; 2Service de Réanimation Polyvalente, Hôpital d’Avignon, 84000 Avignon, France; florent.montini@ch-arles.fr; 3Department of Respiratory Medicine, North Hospital, APHM, 13015 Marseille, France; bermudezjulien@hotmail.fr; 4Aix Marseille University, INSERM, INRAE, C2VN, 13013 Marseille, France; 5Service de Radiologie, Hôpital Nord, APHM, 13015 Marseille, France; mathieu.di-bisceglie@ap-hm.fr

**Keywords:** acute respiratory failure, connective tissue disease, intensive care unit, interstitial lung disease, idiopathic interstitial lung disease

## Abstract

**Background/Objectives**: Interstitial lung disease (ILD) is a heterogenous group of disorders characterised by an association of inflammatory and fibrotic abnormalities of the lung. Acute respiratory failure (ARF) may represent the initial picture of the disease. This study aims to highlight the diagnosis of ILD in the intensive care unit (ICU) and to describe the epidemiological, prognostic, and imaging features of patients diagnosed for the first time with ILD in the ICU. **Methods**: We conducted a single-centre retrospective study. We screened all 2459 patients admitted to our ICU from October 2017 to February 2020. The inclusion criteria consisted of the ILD diagnosis criteria. For each patient, clinical data and lung computed tomography scan patterns were analysed. The selected cases were then reviewed by an expert team at the tertiary care teaching hospital of Marseille (Hôpital Nord, Marseille, France). **Results**: During the study period, 26 ICU patients were diagnosed with ILD and 20 cases were confirmed by the expert team. The most frequent diagnoses were idiopathic ILD (*n* = 7, 35%), auto-immune disease-related ILD (*n* = 7, 35%), exposure-related ILD (*n* = 3, 15%), and carcinomatous lymphangitis (*n* = 3, 15%). Fifteen patients were men (75%), with a mean age of 70 (62–72) years. The median SOFA score was 4 (3–7), and 16 (80%) patients received invasive mechanical ventilation. The mean ratio of the oxygen pressure to the fraction of inspired oxygen was 174 (148–198) mmHg. The ICU mortality rate of our cohort was significantly higher than the average ICU mortality (65% vs. 26%, *p* < 0.003). The mortality rate was lower among the subgroup of auto-immune disease-related ILD (57%). **Conclusions**: We conducted a single-centre cohort study of patients diagnosed with ILD in the ICU. This rare cause of ARF was associated with poor outcome in the ICU, but auto-immune disease-related ILD seemed to have a better prognosis. High-resolution lung CT and identification of lesion patterns are the cornerstones of the diagnosis. Improved knowledge of ILD and multidisciplinary discussion (MDD) involving radiologists, pneumologists, and intensivists may result in an earlier diagnosis and eventually improved treatments.

## 1. Introduction

ILD is a heterogenous group of disorders that cause inflammation and fibrosis in the lungs. Over 200 causes are described, from very rare diseases to more common forms such as idiopathic pulmonary fibrosis, sarcoidosis, and auto-immune diseases. These are listed in an international classification established in 2002 and revised in 2013 [1,2]. ILD occurs in less than 80 patients per 100,000 inhabitants [3]. The pathophysiology of ILD is complex and differs depending on the aetiology. It involves genetic backgrounds, environmental exposure, and auto-immune triggers. It has been demonstrated that it can lead to an acute or chronic inflammation with extracellular matrix and collagen deposition in the pulmonary interstitium, resulting in fibrosis. The initial manifestations of the disease include a progressive cough and dyspnoea, which can progress to chronic respiratory failure in cases of advanced disease. This progression is secondary to impaired gas exchange and loss of lung compliance. However, the presence of ARF can reveal the underlying disease, necessitating admission to an ICU at the onset of the condition. In the context of patients admitted to ICU with ARF attributable to ILD, the mortality rate associated with this condition has been reported to range from 55 to 100% [4]. The treatment of ILD is contingent upon the underlying aetiology and may encompass supplemental oxygen therapy when indicated, physical therapy, and preventive measures, such as vaccines and exposure avoidance strategies [5].

ARF complicating ILD may overlap with the diagnosis of other causes of admission including acute respiratory distress syndrome (ARDS) [6,7]. These patients are considered part of the “diagnoses not typically classified ARDS” and were historically described as “ARDS mimickers” [8]. In a large-scale retrospective study in two French ICUs, the prevalence of ARDS mimickers was 7.5% [9], and most of them had an ILD related to a connective tissue disease, vasculitis, or drug exposure. The management of ARDS mimickers is similar to that of ARDS and is based on symptomatic treatments and ventilator settings in addition to aetiological treatment. Indeed, identifying the cause of ARDS in order to start the treatment at the early phase of the disease before fibrosis develops is critical for the outcome in the ICU [10].

Data on ARF complicated by ILD are scarce, so we undertook our study to (1) estimate the rate of ILD admissions to our ICU; (2) describe the epidemiologic features and outcomes of these patients; and (3) highlight the critical role of an expert team in imaging for the diagnosis of interstitial pneumonia.

## 2. Materials and Methods

We conducted a retrospective single-centre observational study at the ICU of the secondary care general hospital of Avignon, France. The selected cases were reviewed by an expert team in ILD at the tertiary care teaching hospital of Marseille (Hôpital Nord, Assistance Publique Hôpitaux de Marseille, Aix Marseille University, Marseille, France).

From October 2017 to February 2020, we retrospectively included adult patients with ARF due to a newly diagnosed ILD. The diagnosis of ILD was made in accordance with the guidelines established by the American Thoracic Society (ATS) and the European Thoracic Society (ERS), as well as the most recent international updates to these guidelines [1,2,11]. Patients were required to have received invasive or non-invasive mechanical ventilation to be included. A high-resolution lung computed tomography (HRCT) scan was required to confirm the diagnosis. The HRCT was performed within the first 24 h after ICU admission.

Exclusion criteria were ARF due to infection or cardiac failure, patients with an uncertain diagnosis of ILD, and patients with previously diagnosed chronic ILD. The standard exploration at ICU admission was blood testing including a blood fraction count and the C-reactive protein and procalcitonin concentrations. Either a bronchoalveolar lavage (BAL) or a protected distal sampling (PDS) was performed if possible (e.g., if the patient’s trachea was intubated) to exclude a lung or tracheobronchial infection. We evaluated the left cardiac function with point-of-care echocardiography. When the main diagnosis was ILD, patients were selected and analysed for eligibility.

All the survivors were discharged to the department of pneumology of our hospital. Thus, we used the institutional database to assess the medical records. No long-term follow-up data was available. The medical records were assessed in the hospital’s ICU, and then all the data was anonymised. Organ dysfunction was evaluated with the Sepsis-related Organ Failure Assessment (SOFA) score [12].

Patients received at least 72 h of an empirical antibiotic therapy consisting of an association of piperacillin 4 g and tazobactam 0.5 g, four times a day. Fluid management and antibiotic treatment discontinuation or modification were left at the discretion of the patient’s physician according to the French recommendations but were not specifically collected.

We collected data on the demographics and personal medical history of all selected patients. Ventilatory parameters and the mean ratio of the oxygen pressure to the fraction of inspired oxygen (PaO_2_:FiO_2_) were assessed. For patients requiring invasive mechanical ventilation, we also evaluated the pulmonary static compliance and the driving pressure as reported previously [13]. The ICU mortality was also evaluated. For quantitative variables, the worst values available from during the ICU stay were considered.

The medical records and imaging were reviewed by an expert team of pneumologists (J.B.) and radiologists (M.D.B.) who assessed the aetiological diagnosis of ILD as defined by international guidelines [1,2,14,15,16]. The expert team was blinded to the ICU outcome.

Data were expressed as means for quantitative values and percentages for qualitative values, and were compared with Fisher’s exact test, according to the distribution. A *p*-value < 0.05 was considered significant. Statistical analyses were performed using Graph Pad Prism (Software version 10.0.0 for windows, San Diego, CA, USA).

D.E. provided an oral presentation of the primary results of the study during the 2021 French intensive care society congress, which is referenced as follows: “FC-150 Characteristics and outcomes of diffuse interstitial pneumonias discovered in ICU: a retrospective monocentric study (The IPIC study)”.

## 3. Results

Of the 2459 patients admitted to our ICU during the inclusion period, 716 were diagnosed with ARF. The main diagnosis was ILD for 29 patients. After a local review by the intensivist team, three patients met the exclusion criteria: one for mixed cardiogenic and respiratory failure and two for tumoral mass syndrome. The cohort of patients was then reviewed by the expert teams and six other patients were excluded: one patient because of loss of data, one because no HRCT was performed, one for left heart failure, one for infectious pneumonia, and two others for tumoral mass syndrome. After the second expert review, 20 patients with ARF complicated by newly diagnosed ILD were included in our cohort (Figure 1).

The patient’s baseline characteristics and ICU management are listed in Table 1.

Three patients had previously known connective tissue disease (CTD), but none of them had known ILD due to their CTD. Fifteen (75%) were men with a mean age of 70 (62–72). The median SOFA score at ICU admission was 4 (3–7). Most patients (95%) met Berlin’s criteria for ARDS, with a mean PaO_2_:FiO_2_ of 173 (148–198) mmHg.

During the screening time schedule, intensivists successfully diagnosed 77% (20/26) patients with previously unknown ILD (Table 2).

Altogether, among the patients admitted to the ICU during the screening time schedule, 1% of them were confirmed ILD, which represented 3% of all patients admitted with ARF (Table 2).

Table 3 reports the related diseases that cause ILD, as well as the related HRCT patterns that were identified to make the diagnosis.

The main diagnoses were idiopathic ILD (*n* = 7, 35%), auto-immune disease related ILD (*n* = 7, 35%), exposure-related ILD (*n* = 3, 15%) and carcinomatous lymphangitis (*n* = 3, 15%). Exposure-related causes included radiation, the drug docetaxel, and hypersensitivity pneumonitis. Among the three cases of carcinomatous lymphangitis, one patient had a history of thyroid carcinoma. The second case was a patient affected with a gastric adenocarcinoma. The last patient was a woman with a suspicious breast mass with homolateral axillary adenopathy, but she died from her pulmonary condition before further investigation.

The ICU mortality rate was 65%, reaching 81% among those requiring invasive mechanical ventilation (IMV). The IMV survivors had antisynthetase syndrome (anti Jo1 syndrome), exacerbation of idiopathic pulmonary fibrosis, and systemic sclerosis. The ICU mortality rate in our cohort was significantly higher than that of other ICU patients (65% vs. 25.7%, *p* < 0.003) (Table 4).

Four (57%) patients died in the auto-immune disease subgroup.

After discussion at the daily ICU staff meeting, a first-line steroid treatment was administered to six (30%) patients. The steroid infusion consisted of 1 mg/kg of body weight of methylprednisolone for at least three days straight. Once started, the treatment was continued at a high dose until discharge from the ICU. Retrospectively, the aetiology of their ILD was diagnosed as cryptogenic organising pneumonia (COP), hypersensitivity pneumonitis, docetaxel-induced ILD, systemic lupus erythematosus, anti-synthetase syndrome, and systemic sclerosis. The six patients receiving steroids survived their ICU stay.

## 4. Discussion

During the period of study, 1% of our ICU patients were admitted for ILD. In the specific subgroup of patients with ARF, the proportion of ILD was 3%. Our cohort showed a large panel of ILD diagnoses. HRCT was the key element of the diagnosis for the expert team, and identification of the imaging patterns was necessary for the definitive ILD diagnosis.

The most frequent diagnoses were idiopathic ILD (35%), auto-immune-associated ILD (35%), and carcinomatous lymphangitis (15%). The mortality was quite high (65%) with notable variations among two subgroups: mortality was higher in the patients requiring invasive mechanical ventilation (81%) and lower in the patients with auto-immune features (57%). Similar results have been shown previously [4,19]. This emphasises the importance of causal diagnosis in cases of ILD to give the appropriate treatment.

Our study showed that the diagnosis of ILD among ARF patients is infrequent in the ICU but should be considered in a select number of patients. Intensivists should be aware of this disease. They can request the help of an expert team when no clear diagnosis is available during ARF after excluding cardiac failure or an infection. Such data justify performing a thoracic HRCT for patients with ARF when treatments directed against infection or heart failure do not improve the patients and to set an MDD.

To improve diagnosis, BAL should be discussed with patients who can afford it. In our study, these data were unavailable for the patients. However, BAL not only allows clinicians to explore infection, but it can also help inform diagnosis based on cellular patterns and cytology. In a recent study, Chang et al. demonstrated that BAL was useful for diagnosing ILD in the ICU and allowed treatment modification in 60% of patients [20]. However, the benefit of BAL should be balanced by the risks of hypoxia or bronchospasm [21].

In a single-centre study conducted at an ICU of a tertiary centre, Gerard et al. performed surgical lung biopsies on 7% of patients with non-resolving ARDS. This strategy identified 37% of corticosteroid-sensitive pathologies, most of which were idiopathic ILD [22]. Invasive testing, such as a lung biopsy, should be reserved for ILD with an unconfirmed diagnosis, and only after MDD involving radiologists, pulmonologists, and intensivists. However, less invasive methods, such as transbronchial cryobiopsy, can be discussed with the expert team, as they yield diagnoses with a similar degree of confidence (60% vs. 73%) for certain patterns compared to video-assisted thoracoscopic surgery (VATS) [23]. A surgical lung biopsy should be discussed as a last resort in MDD since it is associated with a higher mortality risk in these patients, with a 30-day post-procedure mortality rate of around 5% [5].

Early identification of ILD in patients with ARF in the ICU could lead to rapid treatment of the underlying disease. In our cohort, steroid administration was the only specific therapy used. Recommendations maintain steroids as the first-line treatment for acute exacerbation of idiopathic lung disease [15,23], and they are often mentioned with good results in specific diagnoses such as organising pneumonia [24]. However, recent data suggest favourable results with antifibrotic and immunomodulatory therapies for various diagnoses [23]. Indeed, patients with ILD associated with anti-MDA-5 antibodies (aMDA-5) have a high 6-month mortality rate exceeding 50%, but they may have a better prognosis. A combination of steroids, calcineurin inhibitors, and cyclophosphamide (CYC) has been shown to significantly improve outcomes for these patients [25]. Another team has also shown improved survival in a recent series of patients using a combination of immunosuppressive drugs, including JAK inhibitors [26]. However, in patients with exacerbation of idiopathic pulmonary fibrosis, CYC use has recently been shown to be associated with higher mortality, suggesting that immunosuppressive therapy should be avoided for such patients [27]. Rituximab treatment in conjunction with mycophenolate mofetil (MMF) has also been shown to be more effective than MMF alone in patients with non-specific interstitial pneumonia (NSIP) [28]. Finally, nintedanib, an antifibrotic agent known for its efficacy in slowly progressive fibrotic ILD, may be a new therapeutic option for acute exacerbations of fibrosing ILD, as Urushiyama et al. recently demonstrated [29].

Taken together, these data underscore the importance of discussing ILD diagnoses with an expert team after excluding infection and heart failure to provide optimal treatment for patients.

More knowledge about ILD in the ICU could also be necessary to justify therapeutic limitations. For example, patients with ILD associated with aMDA-5 or patients with acute exacerbation of idiopathic pulmonary fibrosis requiring invasive mechanical ventilation have, respectively, 84% and 87% mortality in the ICU [30,31]. Our results are consistent with these findings. Of note, our only patient with aMDA-5 died. On the other hand, organising pneumonia is known to have a good prognosis when treated with steroids [32]. In our cohort, the only patient affected with organising pneumonia survived after steroid treatment.

Thus, intensivists need to know more about ILD and discuss or integrate an expert team to define the best care for patients.

Our study specifically focused on patients with previously unknown ILD. Dhanani et al. recently published a comprehensive review of the management of ILD patients in the ICU, including those with acute exacerbation of previously diagnosed ILD [33]. This review summarises the most recent knowledge and could be considered a reference, as there is no standard of care for ILD in the ICU. Since our study was not prospective, the intensivists did not follow any specific therapeutic bundle to manage ILD patients. However, we can see that the ventilatory management of ILD in the ICU was applied according to the latest data, which implies a wide range of oxygen and ventilation strategies, such as high-flow nasal oxygen (HFNO), mechanical ventilation with low positive end-expiratory pressure (PEP), and, if necessary, veno-venous extracorporeal membrane oxygenation (ECMO) (see Table 1).

We must acknowledge several limitations. First, our study was monocentric and retrospective, which could have introduced biases. We lack therefore potentially interesting data, as non-invasive ventilation, duration of corticosteroid treatment, microbiological results, or the type of antibiotic regimens. Other data were not available because they are not included in the standard care in our centre (for example VATS or cryobiopsy). Other limitations are the retrospective design and the treatments left to the physician’s discretion. Furthermore, this study was conducted at a single centre, which limits the number of patients for this rare disease. Only 20 patients fulfilled the eligibility criteria over the period of inclusion (29 months), which is a very limited sample, while no long-term follow-up was collected. Therefore, the generalisation of the results should be approached with caution. Nevertheless, few studies have evaluated acute and previously unknown ILD in the ICU, and our study highlights the importance of knowing about such diseases for intensivists.

## 5. Conclusions

Over the period of study, 20 ICU patients were successfully diagnosed with previously unknown ILD. The ICU mortality was higher than the average ICU mortality during the inclusion period. HRCT is critical for diagnosing ILD among patients with ARF. A multidisciplinary approach including an expert radiology team and pneumologists is essential to confirm the diagnosis of ILD and possibly start a specific treatment, but intensivists should be familiar with this disease as first-line provider.

## Figures and Tables

**Figure 1 diagnostics-15-01995-f001:**
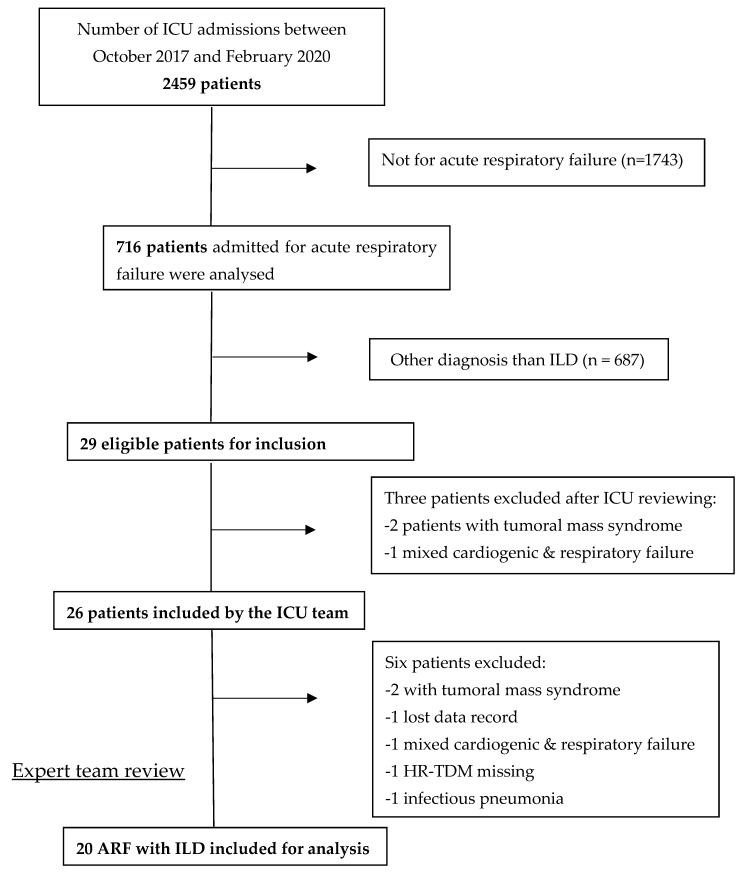
Flow chart of patients screened from October 2017 to February 2020.

**Table 1 diagnostics-15-01995-t001:** Baseline characteristics and ICU management.

Demographics	Median (IQR) or Mean (%)
Age	70 (62–72)
Women	5 (25)
Current smoker	4 (20)
Allergic rhinitis	1 (5)
Cardiovascular disease	6 (30)
Obese	2 (10)
Known cured or evolutive cancer disease	5 (25)
Connective tissue disease	3 (15)
Systemic sclerosis	1 (5)
Rheumatoid arthritis	1 (5)
Systemic lupus erythematosus	1 (5)
**Severity**	
SOFA Score	4 (3–7)
**Criteria for ARDS**	19 (95)
Mild	5 (25)
Moderate	11 (55)
Severe	3 (15)
**Ventilatory support**	
HFNO	3 (15)
Invasive ventilation	16 (80)
Veno-venous ECMO	1 (5)
**Ventilatory parameters**	
PaO_2_:FiO_2_	174 (148–198)
PEP (cmH_2_O)	8 (8–10)
Driving pressure (cmH_2_0)	19 (17–20)
Static compliance (mL/cmH_2_0)	21 (18–24)

ECMO: extracorporeal membrane oxygenation; HFNO: high-flow nasal oxygen; PaO_2_:FiO_2_: ratio of the oxygen pressure to the fraction of inspired oxygen; PEP: positive end-expiratory pressure; SOFA: Sepsis-related Organ Failure Assessment; respiratory system static compliance (Crs): measured by the formula Crs = Vt/DP, where Cr is the static compliance of the respiratory system, Vt is the tidal volume in assist-control ventilation, and DP is the driving pressure. Driving pressure (DP): estimated by DP = Plateau pressure–total PEP, where DP is the driving pressure and Pplat is the plateau pressure measured after an inspiratory pause. The total PEP is the pressure measured after a tele-expiratory pause on a ventilator on assist-control ventilation. See [17].

**Table 2 diagnostics-15-01995-t002:** Incidence of ILD in the ICU.

Total ICU Admission	2459
ILD initially diagnosed by the ICU team	26
Confirmed ILD among ICU team’s selection (%)	20/26 (77)
Confirmed ILD among all ICU admissions (%)	20/2459 (~1)
Confirmed ILD among ARF patients (%)	20/687 (~3)

**Table 3 diagnostics-15-01995-t003:** Main diagnosis and associated HRCT pattern.

ILD	Related Main HRCT Pattern	*n* = 20 (%)
**Idiopathic ILD**		**7 (35)**
Idiopathic non-specific interstitial pneumonia	NSIP	3 (15)
Cryptogenic organising pneumonia	OP	1 (5)
Idiopathic pulmonary fibrosis	UIP	3 (15)
**Auto-immune-related ILD**		**7 (35)**
CTD associated ILD		6 (30)
Systemic lupus erythematosus	Shrinking lung syndrome	1 (5)
Rheumatoid arthritis	UIP	1 (5)
Systemic sclerosis	NSIP	2 (10)
Anti-synthetase syndrome (anti-Jo1 syndrome)	NSIP	1 (5)
aMDA-5-associated amyopathic dermatomyositis	NSIP	1 (5)
Interstitial pneumonia with auto-immune features	NSIP	1 (5)
**Exposure-related**		**3 (15)**
Hypersensitivity pneumonitis	HP	1 (5)
Drug-induced (docetaxel)	NSIP	1 (5)
Radiation-induced lung injury	NSIP	1 (5)
**Others**		**3 (15)**
Carcinomatous lymphangitis	Crazy paving	3 (15)

aMDA-5: anti-MDA-5 antibodies. **Imaging pattern**: CTD: connective tissue disease; UIP: usual interstitial pneumonia; NSIP: non-specific interstitial pneumonia; OP: organising pneumonia; HP: hypersensitivity pneumonitis (for the HRCT pattern description, see [14]). Shrinking lung syndrome: see [18].

**Table 4 diagnostics-15-01995-t004:** Comparison of ICU outcomes among admissions between October 2017 and February 2020.

	ILD Patient with IMV	Patient with Auto-Immune-Related ILD	ILD Patient with Steroid Administration
Number of patients	16	7	6
Deceased in the ICU	13	4	0
Mortality rate (%)	81	57	0
	**Confirmed ILD**	**Overall ICU patients**	** *p* ** **-value**
Number of patients	20	2459	
Deceased in the ICU	13	632	
Mortality rate (%)	65	26	*<0.003*

ICU: intensive care unit; ILD: interstitial lung disease; IMV: invasive mechanical ventilation.

## Data Availability

The datasets used and/or related materials that support the findings of this study are not publicly available due to French law restrictions because they contain information that could compromise the privacy of research participants. However, these are available from D.E. on reasonable request.

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
