# Peer review of "Characteristics and Outcomes of Diffuse Interstitial Pneumonias Discovered in the ICU: A Retrospective Monocentric Study—The “IPIC” (Interstitial Pneumonia in Intensive Care) Study"

_diagnostics, 2025, doi:10.3390/diagnostics15161995_

Round 1

Reviewer 1 Report

Comments and Suggestions for Authors

Thank you for the opportunity to review the manuscript. It is focussed on an interesting subject in clinical ILD care.

The outcome of ILD exacerbation requiring ICU therapy is a crucial clinical problem under ongoing discussion. Only few data are available.

Although the current manuscript is based on a monocentric retrospective data analysis, it is of particular clinical interest concerning frequency of newly diagnosed ILD in ICU admission due to ARF.

However there are several issues that need to be raised

  • The authors should clearly state that ILD was firstly diagnosed while admitted to the ICU due to ARF.
  • There is a discrepancy between table 1 and table 3 concerning frequency of CTD- ILD
  • The authors should state when the CT has been performed (at admission etc.) and which other diagnostic measures has been used, especially to diagnose infectious conditions.
  • Table 3 is difficult to read:

iNSIP comes as cellular and fibrotic form that should be distinguished,

Shrinking lung syndrome is not an CT pattern but a clinical condition and per se not an ILD. Did you check diaphragm mobility ?

What kind of Anti synthetase syndrome has been diagnosed ?

HP is a diagnosis but not an CT pattern: was it UIP, NSIP, head cheese sign etc. ?

What kind of cancer were diagnosed in patients with carcinomatous lymphangitis ?

Did any patients receive NIV and how many had NIV failure ? How long was the stay at the ICU ?

Which 4 patients did survive IMV ?

Secondly, the treatment regimen (antibiotics, fluid management etc.) should be described if possible.  

About the steroid therapy: 6 patients received methylpred for 3 days. How where these patients choosen, and what was the reason not to treat the other patients?

Was the steroid therapy stopped after 3 days in the 6 patients ? If not, how  was the treatment continued ?

Author Response

Dear reviewer,

Thank you for the time you have devoted to our article. As requested, you will find bellow

the answers to the various points raised by your reviewing.

The main text has been modified considering the remarks of both you and the second
reviewer.

Sincerely yours,

Damien Eckert and Marc Leone

  • The authors should clearly state that ILD was firstly diagnosed while admitted to the ICU due to ARF.

>>> We modified the introduction as follow:

with ILD diagnosed in our ICU department.

Has been changed for

patients with ILD firstly diagnosed in the ICU department.

Also in the exclusion criteria you can find that, as quoted here:

“patients with previously diagnosed chronic ILD” were excluded from the cohort.

  • There is a discrepancy between table 1 and table 3 concerning frequency of CTD- ILD

Table 1 states the historical records from the patients at baseline admission and does not include the ILD diagnosis. As explain in the main text just below:

“Three patients had previously known connective tissue disease (CTD), but none of them had known ILD due to their CTD.”

  • The authors should state when the CT has been performed (at admission etc.) and which other diagnostic measures has been used, especially to diagnose infectious conditions.

In the subheading “Material and Methods” we have added this sentence:

The HRCT was performed within the first 24 hours of ICU admission.

And a few sentences later:

“The standard exploration at the admission of every patient was a blood testing with a blood fraction count, C-reactive protein and Procalcitonin levels. Either a Bronchoalveolar lavage (BAL) or a protected distal sampling (PDS) was performed if possible (e.g if the patient trachea was intubated) to exclude a lung or tracheobronchial infection. We evaluated the left cardiac function with a point-of-care echocardiography.”

  • Table 3 is difficult to read:
  • iNSIP comes as cellular and fibrotic form that should be distinguished
    Thank you for your relevant comment. It is indeed an histological definition, and since we did not have a pulmonary biopsy, we were unable to discriminate the two forms of NSIP. We thus let the mention “NSIP” in the table in both cases.

  • Shrinking lung syndrome is not an CT pattern but a clinical condition and per se not an ILD. Did you check diaphragm mobility?
    Shrinking lung syndrome is known presentation of systemic lupus erythematosus defined by small lung, atelectasis, with diaphragmatic elevation (as highlighted by Duron L, Cohen-Aubart F, Diot E, Borie R, Abad S, Richez C, Banse C, Vittecoq O, Saadoun D, Haroche J, Amoura Z. Shrinking lung syndrome associated with systemic lupus erythematosus: A multicenter collaborative study of 15 new cases and a review of the 155 cases in the literature focusing on treatment response and long-term outcomes. Autoimmun Rev. 2016 Oct;15(10):994-1000. doi: 10.1016/j.autrev.2016.07.021..). The mobility is not part of the definition diagnosis. To avoid a mistake on the diagnosis, all cases were discussed in a multidisciplinary meeting including an intensivist, a senior radiologist and a senior pneumologist. Thank to your comment, we decided to include the reference mentioned above to clarify the table 3 as you will see in the main text.

  • What kind of Anti synthetase syndrome has been diagnosed?
    >>> It was an anti-Jo1 syndrome, we added this type of antisynthetase syndrome in the table 3. It has been added as well in main text.

  • HP is a diagnosis but not an CT pattern: was it UIP, NSIP, head cheese sign etc. ?
    We adhered to the Official ATS/JRS/ALAT Clinical Practice Guideline of 2020 (Raghu G, Remy-Jardin M, Ryerson CJ, Myers JL, Kreuter M, Vasakova M, Bargagli E, Chung JH, Collins BF, Bendstrup E, Chami HA, Chua AT, Corte TJ, Dalphin JC, Danoff SK, Diaz-Mendoza J, Duggal A, Egashira R, Ewing T, Gulati M, Inoue Y, Jenkins AR, Johannson KA, Johkoh T, Tamae-Kakazu M, Kitaichi M, Knight SL, Koschel D, Lederer DJ, Mageto Y, Maier LA, Matiz C, Morell F, Nicholson AG, Patolia S, Pereira CA, Renzoni EA, Salisbury ML, Selman M, Walsh SLF, Wuyts WA, Wilson KC. Diagnosis of Hypersensitivity Pneumonitis in Adults. An Official ATS/JRS/ALAT Clinical Practice Guideline. Am J Respir Crit Care Med. 2020 Aug 1;202(3):e36-e69. doi: 10.1164/rccm.202005-2032ST. Erratum in: Am J Respir Crit Care Med. 2021 Jan 1;203(1):150-151. doi: 10.1164/rccm.v203erratum1. Erratum in: Am J Respir Crit Care Med. 2022 Aug 15;206(4):518. doi: 10.1164/rccm.v206erratum4. PMID: 32706311; PMCID: PMC7397797.) and add it in the method.

This guideline reported that an HRCT pattern has been described for typical Hypersensitivity Pneumonitis as a pattern in itself, which we chose to be more accurate.

Thank you again for this comment, as we added the references in the cited articles.

  • What kind of cancer were diagnosed in patients with carcinomatous lymphangitis ?

Among the three cases of carcinomatous lymphangitis, one patient was affected with a known thyroid carcinoma. The second case was a patient affected with a gastric adenocarcinoma. The last patient was a women with a suspicious breast mass with homolateral axillary adenopathy, but she died from her pulmonary condition before further investigation.

We added this detail in the manuscript.

Did any patients receive NIV and how many had NIV failure? How long was the stay at the ICU ?

The retrospective design does not allow to respond to your question. We have included a section in the discussion to explain why this data is lacking, and how it limits the conclusion of our study.

Which 4 patients did survive IMV?

We were very confused on that matter, because there was a misprint in the table 4. Only three patients survive IMV in our cohort:
-one with exacerbation of idiopathic pulmonary fibrosis
-the patient affected with Syndrome anti-synthetase syndrome (Jo1)
-The patient affected with Systemic sclerosis.

We corrected the mistake and added this information in the main text.

Secondly, the treatment regimen (antibiotics, fluid management etc.) should be described if possible.  

To respond to your interesting comment, we added on the main text information about your request on the material and method subheading as you will see:

Patients received at least 72 hours of an empiric antibiotic therapy consisting of an association of piperacillin 4 g and tazobactam 0.5 g, four times a day. Fluid management and antibiotic treatment discontinuation or modification were let at the discretion of the patient’s physician according to the French recommendations but were not specifically collected.

About the steroid therapy: 6 patients received methylpred for 3 days. How where these patients choosen, and what was the reason not to treat the other patients?

The treatment has been decided after discussion at the daily intensive care round and was focussed on patients with auto-immune affections. Once started, the treatment was continued at high dose until discharge from the ICU.

Was the steroid therapy stopped after 3 days in the 6 patients ? If not, how  was the treatment continued ?

Once started the treatment was continued until discharge from the ICU (or death).

Reviewer 2 Report

Comments and Suggestions for Authors

Dear authors,

Major points

Study size and power – Only 20 patients fulfilled your eligibility criteria over 29 months (≈ 1 % of all ICU admissions, 3 % of ARF cases) .  This limited sample precludes any meaningful multivariable analysis of prognostic factors and yields extremely wide confidence intervals for reported proportions (e.g. mortality).  Please add a post-hoc power calculation or, preferably, temper the strength of causal language throughout the Discussion.

Case ascertainment and selection bias – You excluded nine of 29 initially eligible patients after two rounds of internal review (Fig. 1) , yet do not detail the decision rules applied by the “expert team”.  Provide explicit diagnostic criteria for each ILD category (UIP, NSIP, HP, etc.) and clarify whether reviewers were blinded to outcomes.  Otherwise the reader cannot judge reproducibility.

Statistical methods – The text states that continuous variables were compared with the Fisher exact test , which is reserved for categorical data.  Re-specify the tests actually used (e.g. Mann–Whitney U, Student t) and confirm that normality and variance assumptions were checked.

Definition of ARF/ARDS – Nineteen patients “met Berlin criteria for ARDS” , yet the manuscript does not show how timing, imaging and PEEP thresholds were verified.  Add these details or a supplementary table documenting the diagnostic components for each case.

Therapeutic interventions – Six patients received high-dose steroids and all survived , but the indication, timing, and taper are not specified.  Similarly, only one patient underwent ECMO support.  Readers need a clearer description of decision algorithms for steroids, immunosuppressants and ECMO to interpret the outcome differences you highlight.

Missing data on BAL / histology – You acknowledge that BAL results were unavailable ; however, the absence of any microbiological work-up leaves residual confounding by infection.  Please report how many patients underwent BAL, trans-bronchial cryobiopsy or VATS, and summarise key cytology or culture findings where they exist.

Long-term outcomes – ICU survival is important but insufficient.  Provide in-hospital and 90-day mortality, ventilator-free days and disposition (home vs. rehabilitation) if available, or explicitly state that no follow-up data were collected.

Figures and tables – Table 4 mixes exposure variables (ventilatory modes) with outcomes (mortality) and contains unexplained abbreviations (“PEP”) .  Re-organise tables into baseline characteristics, ICU management and outcomes, and define every abbreviation in footnotes.

Minor points

Language and style – A careful copy-edit is required to correct typographical errors (“fraction of inspired oxygen ratio” should be “fraction of inspired oxygen” etc.) and to harmonise abbreviations (first mention of ARF, HRCT, ILD, etc.).

Redundancy – Several sentences repeat background information already given (e.g., lines 45–63 vs. 219–236).  Streamline the Introduction and Discussion.

Reference formatting – MDPI style requires author initials before surnames and inclusion of DOI for journal articles.  Please revise the reference list accordingly.

Ethics paragraph – Move the Institutional Review Board approval and consent statements from the Methods to a standalone “Ethics approval” subsection, following journal guidance.

Spelling of acronyms – Use standard British spelling (“programme”, “randomised”) unless quoting guideline titles, and ensure consistency (e.g., “connective tissue disease” not “connective-associated”).

Comments on the Quality of English Language

The manuscript requires moderate revision for English language quality. While the overall meaning is clear, there are numerous grammatical errors, awkward phrasings, and inconsistent use of abbreviations and terminology. A professional copy-edit is recommended to correct typographical mistakes, improve sentence structure, and ensure clarity and consistency throughout the text.

Author Response

Dear reviewer,

Thank you for the time you have devoted to our article. As requested, you will find bellow

the answers to the various points raised by your reviewing.

The main text has been modified considering the remarks of both you and the first
reviewer.

Sincerely yours,

Damien Eckert and Marc Leone

Major points

Study size and power – Only 20 patients fulfilled your eligibility criteria over 29 months (≈ 1 % of all ICU admissions, 3 % of ARF cases).  This limited sample precludes any meaningful multivariable analysis of prognostic factors and yields extremely wide confidence intervals for reported proportions (e.g. mortality).  Please add a post-hoc power calculation or, preferably, temper the strength of causal language throughout the Discussion.

>>> Thank you for this comment, we tempered the results in the discussion.

Case ascertainment and selection bias – You excluded nine of 29 initially eligible patients after two rounds of internal review (Fig. 1), yet do not detail the decision rules applied by the “expert team”.  Provide explicit diagnostic criteria for each ILD category (UIP, NSIP, HP, etc.) and clarify whether reviewers were blinded to outcomes. Otherwise the reader cannot judge reproducibility.

Thank you for this comment, we have added every  ATS recommendations that were used to exclude the cases throughout the second row of reviewing (during the MDD).

Statistical methods – The text states that continuous variables were compared with the Fisher exact test , which is reserved for categorical data.  Re-specify the tests actually used (e.g. Mann–Whitney U, Student t) and confirm that normality and variance assumptions were checked.

We have only used the Fisher exact test for categorial variable (deceased or survivor) according to the main affection of the ICU admission (ILD or non-ILD).

Definition of ARF/ARDS – Nineteen patients “met Berlin criteria for ARDS”, yet the manuscript does not show how timing, imaging and PEEP thresholds were verified.  Add these details or a supplementary table documenting the diagnostic components for each case.
>>> This data is regrettably not available as the data were collected retrospectively.

Therapeutic interventions – Six patients received high-dose steroids and all survived, but the indication, timing, and taper are not specified.  Similarly, only one patient underwent ECMO support.  Readers need a clearer description of decision algorithms for steroids, immunosuppressants and ECMO to interpret the outcome differences you highlight.

Thank you for this comment. We have added in the text that since the study was retrospective, there were no prospective therapeutic bundle or algorithms that were used, and that therapeutic decisions were left at the discretion of in charge physicians. 
Regarding methylprednisolone you can now read this section in the results subheading:

“After discussion during the ICU staff meeting, a steroid first-line treatment was administered to 6 (30%) patients. The steroid infusion consisted of 1 mg/kg of body weight of methylprednisolone for at least three days straight. Once started this therapy was continued until the patient was discharged from the ICU. Retrospectively, the aetiology of their ILD was diagnosed as cryptogenic organizing pneumonia (COP), hypersensitivity pneumonitis, docetaxel-induced ILD, systemic lupus erythematosus, anti-synthetase syndrome, and systemic sclerosis. The 6 patients receiving steroids survived to the ICU stay.

Missing data on BAL / histology – You acknowledge that BAL results were unavailable ; however, the absence of any microbiological work-up leaves residual confounding by infection.  Please report how many patients underwent BAL, trans-bronchial cryobiopsy or VATS, and summarise key cytology or culture findings where they exist.

We have considered and modified the text in several areas to answer this comment. There were no data on histology since we lacked less invasive methods, such as transbronchial cryobiopsy or VATS in the hospital of Avignon.
Patients without invasive mechanical intubation did not undergo BAL, and for others the bacterial culture was negative (because an infectious affection meets the exclusion criteria).
Only few patients had BAL with cytology exploration because it was part of the intensivists standard exploration during the study period.
Of note, the patient affected with HP had a BAL with lymphocytosis 21% which was helpful to make the diagnosis, but as it is an isolated case of our cohort, we thus decided not to mention it.

Long-term outcomes – ICU survival is important but insufficient.  Provide in-hospital and 90-day mortality, ventilator-free days and disposition (home vs. rehabilitation) if available, or explicitly state that no follow-up data were collected.
We have stated that no long-term follow-up data was collected and acknowledged this limitation in a dedicated paragraph.

Figures and tables – Table 4 mixes exposure variables (ventilatory modes) with outcomes (mortality) and contains unexplained abbreviations (“PEP”) .  Re-organise tables into baseline characteristics, ICU management and outcomes, and define every abbreviation in footnotes.

We have deleted the original table 4 and pushed the related information into a new and extended table 1 (Baseline). The table 4, new version, states mortality among subgroups and also the compared ILD patients/overall mortality.

Minor points

Language and style – A careful copy-edit is required to correct typographical errors (“fraction of inspired oxygen ratio” should be “fraction of inspired oxygen” etc.) and to harmonise abbreviations (first mention of ARF, HRCT, ILD, etc.).

>>> Done

Redundancy – Several sentences repeat background information already given (e.g., lines 45–63 vs. 219–236).  Streamline the Introduction and Discussion.

>>> The lines you are referring to are not the same than the main text I uploaded so that I cannot understand where the redundancy takes place.

Reference formatting – MDPI style requires author initials before surnames and inclusion of DOI for journal articles.  Please revise the reference list accordingly.

>>> Done

Ethics paragraph – Move the Institutional Review Board approval and consent statements from the Methods to a standalone “Ethics approval” subsection, following journal guidance.

>>> Done

Spelling of acronyms – Use standard British spelling (“programme”, “randomised”) unless quoting guideline titles, and ensure consistency (e.g., “connective tissue disease” not “connective-associated”).

>>> Done

Round 2

Reviewer 2 Report

Comments and Suggestions for Authors

Dear authors,

The revised manuscript is substantially improved. The authors have clarified inclusion/exclusion criteria, specified that HRCT was performed within 24 h of ICU admission, described the empirical antibiotic regimen, and expanded the limitations paragraph, all of which directly address the main concerns raised for version 1. The scientific message remains clinically relevant, and the additional methodological detail strengthens reproducibility.

Minor points

Typographical consistency: insert a space in “4 [3-7] and” (Results), correct “charac-terised” in the Abstract, and ensure uniform use of en-dashes for ranges.

Abbreviations: define BAL, PDS and MDD at first appearance; use either “aMDA-5” or “anti-MDA5” consistently.

Reference style: duplicates ATS/ERS 2002 vs. ATS on line 360 – check reference numbering.

Table 1 header could read “Baseline characteristics and ICU management” (currently two titles overlap).

Author Response

Dear reviewers,

thank you for your very positive feedback on our last version of the manuscript.

You will find attached the manuscript in its second revision after the minor points that had to be ameliorate.

Best regards,

Damien Eckert and Marc Leone.

Typographical consistency: insert a space in “4 [3-7] and” (Results), correct “charac-terised” in the Abstract, and ensure uniform use of en-dashes for ranges.

>Done

Abbreviations: define BAL, PDS and MDD at first appearance; use either “aMDA-5” or “anti-MDA5” consistently.

>Done

Reference style: duplicates ATS/ERS 2002 vs. ATS on line 360 – check reference numbering.

>Done

Table 1 header could read “Baseline characteristics and ICU management” (currently two titles overlap).

>Done